# What is the mission of innovation?—Lexical structure, sentiment analysis, and cosine similarity of mission statements of research-knowledge intensive institutions

**Julián D. Cortés**[1,2,3]*

**1** School of Management and Business, Universidad del Rosario, Bogotá, Colombia, **2** Fudan Development Institute, Fudan University, Shanghai, China, **3** School of Business, Woxsen University, Telangana, India

* julian.cortess@urosario.edu.co

**Data Availability Statement:** The following permanent link provides access to the missions sourced, thereby facilitating further replications/triangulations: https://doi.org/10.34848/ZTL686.

## Abstract

Mission statements (henceforth: missions) are strategic planning communication tools used by all types of organizations worldwide. Missions communicate an organization's purpose, values, standards, and strategy. Research on missions has been prolific over the past 30 years, nevertheless several empirical gaps remain, such as single sector or country studies and restricted mission samples. In this article, we identify similarities and differences in the content of missions from government, private, higher education, and health research-knowledge intensive institutions in a sample of 1,900+ institutions from 89 countries through the deployment of sentiment analysis, readability, and lexical diversity; semantic networks; and a similarity computation between document corpus. We found that missions of research-knowledge intensive institutions are challenging to read texts with lower lexical diversity that favors positive rather than negative words. In stark contrast to this, the non-profit sector is consonant in multiple dimensions in its use of Corporate Social Responsibility jargon. The lexical appearance of 'research' in the missions varies according to mission sectorial context, and each sector has a cluster-specific focus. Utilizing the mission as a strategic planning tool in higher-income regions might serve to explain corpora similarities shared by sectors and continents. Furthermore, our open-access dataset on missions worldwide can be used as a source for further replication, triangulation, or crowdsourcing-data studies. Also, practitioners could use our open-access dataset and insights to facilitate strategic planning activities in organizations from multiple sectors.

## Introduction

A recent article in *The Economist* discussed the meaning of mission statements—henceforth mission(s)—and what stakeholders can learn from how organizations describe their goals and priorities [1]. Let us begin by considering three examples of missions from different sectors:

**Funding:** JDC Research grant IMA Research Foundation https://www.imanet.org/educators/research-foundation?ssopc=1 The funders had no role in study design, data collection and analysis, decision to publish, or preparation of the manuscript.

**Competing interests:** The authors have declared that no competing interests exist.

- Higher education—MIT: 'The mission of MIT is to advance knowledge and educate students in science, technology, and other areas of scholarship that will best serve the nation and the world in the 21st century.'

- Government—Council of Scientific and Industrial Research, India: 'CSIR's renewed mission is inspired by the remarks made by [the] President of CSIR Society to CSIR to build "the new CSIR that will fulfill the aspirations of modern India." So CSIR's mission is simply–to build a new CSIR for a new India.'

- Private—Google Inc.: 'Organize the world's information and make it universally accessible and useful.'

Through missions, organizations communicate their purpose, values, standards, and strategy [2]. Research on missions has been prolific over the past 30 years. A recent literature review on the subject states that the main focus of this research is threefold [3]: i) comparative performance of firms that report their mission publicly versus those that do not; ii) the specific characteristics of mission content and orientation; iii) and the mediating effect of diverse factors (e.g., employees' commitment to the mission) on both financial (i.e., observable and self-reported) and non-financial performance. To date, most studies have analyzed institutions, ranging from private firms to hospitals and universities, solely from higher-income countries [4–21].

While acknowledging the substantial insights into the field of strategic planning management produced by the research agenda on missions, we argue that there are three areas in which the research agenda could be enhanced: i) since most of the studies have focused on private single-sector institutions in a single country, the study sample could examine cross-national and multi-sector institutions; ii) since most of the institutions are in the service and manufacturing sectors, knowledge-intensive institutions could also be examined; and iii) since the mission focus has hitherto been the firm (i.e., private sector), missions from governmental institutions could also be included.

It is important to point out that this study is not concerned with providing a new or complementary definition of innovation or any of its derivates (e.g., social-open innovation) through content analysis, provision of which can be found elsewhere [22]. The specific purpose of our study is to identify similarities and differences in the content of missions from government, private, higher education, and health research-knowledge intensive institutions in a sample of 1,900+ entities from 89 countries. The study is guided by the following research questions (RQs):

- RQ$_1$: What are the content and its structure of the missions of research-knowledge intensive institutions?

- RQ$_2$: Are there similarities between the content and its structure of government and non-government missions?

- RQ$_3$: Are mission content similarities identifiable across sectors and continents?

The study is divided into four sections. After this introduction, the second section presents the materials and methodology. In the third section the results are presented. And in the final section we discuss the results in conjunction with current literature on the topic and conclude the study with an assessment of its restrictions and suggestions for future research.

## Materials and methodology

This section presents the data, and the sampling followed to avoid any bias towards the sample's most representative type of institution: higher education. This is followed by a brief

reference to methodological appraisals, the specific details of which will be presented in the subsequent section. Lastly, we mention the software and packages used.

## Data and sampling

We used as a baseline the SCImago Institutions Rankings (SIR) [23]. The SIR ranks over 5,000 institutions worldwide from five sectors: government (e.g., NASA, US Army, NOOA), health (e.g., NIH, National Library of Medicine, National Institute of Biomedical Imaging and Bioengineering), higher education (e.g., Stanford University, MIT, Yale University), private (e.g., Google, Pfizer, Ford Motors), and non-profit (e.g., Qatar Foundation, National Bureau of Economic Research, or the Bill and Melinda Gates Foundation). Institutions included in the ranking need to have published at least 100 documents indexed in Scopus [24]. SIR also takes into account the research impact of newly published patents and of academic publications cited in those patents and further calculates the societal impact of an institution based on website incidence. The indicator is composed of the following variables:

- 50% of the score corresponds to a research component: output, international collaboration, normalized impact, high-quality publications, excellence, scientific leadership, leadership excellence, and scientific talent pool.

- 30% corresponds to the innovation component: innovation knowledge, patents, and technological impact.

- The remaining 20% corresponds to the societal impact component: websites and inbound links.

The SIR was recently updated and includes aggregated new variables. The missions in our study sample, however, were sourced before the SIR was updated.

We considered only those institutions with a consistent presence in the SIR from 2014 to 2021. These institutions show that despite individual rank, research-innovation performance is a central activity and one that requires a stable environment for the institutions to function—at least in the mid-term.

We sourced the missions manually from the official institutional websites. We also considered other related categories such as *organization*'s *description; introduction; about us; our purpose; our values; our goals; our objective*; and *what we do*, among others. The sample analyzed was restricted to missions in English, due to the limitations of the automated analysis tools, and comprises 1,955 institutions from 89 countries. Table 1 presents the sample's composition by sector and continent.

Most of the missions were sourced from higher education and health institutions, followed by government, private, and 'Others' (i.e., non-profit) institutions. Due to the sample's bias towards higher education institutions, we will use the following (sub)samples to conduct the analyses:

- We used the complete sample to conduct an exploration of the missions' content (*n = 1955*—Table 1). The aim of this exploration is to identify the mission content features in terms of readability, diversity, and sentiment analysis, and is limited to producing a descriptive and correlational glimpse. Caution should be exercised in interpreting our findings given the sample's bias towards higher education institutions.

- Semantic networks are used to produce a structural examination of the content of the missions. We will assemble the semantic networks as follows:

    - All sectors (the general mission-driven innovation): a stratified sample will be implemented. Since the 'Others' category is the least numerous with 24 missions, 24 randomly

**Table 1. Mission composition by sector and continent.**

| Sector/Continent | # Institutions | % |
|---|---|---|
| Government | 300 | 15,35 |
| Africa | 7 | 0,36 |
| Asia | 103 | 5,27 |
| Europe | 133 | 6,80 |
| LATAM-CAR | 2 | 0,10 |
| North America | 49 | 2,51 |
| Oceania | 6 | 0,31 |
| Health | 337 | 17,24 |
| Africa | 5 | 0,26 |
| Asia | 59 | 3,02 |
| Europe | 93 | 4,76 |
| LATAM-CAR | 1 | 0,05 |
| North America | 150 | 7,67 |
| Oceania | 29 | 1,48 |
| Higher ed. | 1261 | 64,50 |
| Africa | 55 | 2,81 |
| Asia | 451 | 23,07 |
| Europe | 361 | 18,47 |
| LATAM-CAR | 11 | 0,56 |
| North America | 351 | 17,95 |
| Oceania | 32 | 1,64 |
| Others | 24 | 1,23 |
| Asia | 1 | 0,05 |
| Europe | 5 | 0,26 |
| North America | 18 | 0,92 |
| Private | 33 | 1,69 |
| Asia | 6 | 0,31 |
| Europe | 9 | 0,46 |
| North America | 17 | 0,87 |
| Oceania | 1 | 0,05 |
| Total general | 1955 | 100,00 |

Source: the authors' work, based on the organizations' website. Note: LATAM-CAR is Latin America and the Caribbean.

chosen missions will be sourced from the remaining categories. This will ensure that the same number of missions for each sector is analyzed. (*n = 120*—Table 2).

- Government institutions (the governmental mission-driven innovation): the complete category has 300 missions. All missions will be considered since, to the best of our knowledge, this will be the first study to produce a semantic network of knowledge-research intensive institutions in the government sector. (*n = 300*—Table 1). The importance of an in-depth study of governmental research-knowledge institutions and of contrasting their missions with those of other sectors lies the mission's pivotal role in promoting and upgrading domestic industries, creating new markets, and supporting research for health breakthroughs [25, 26].

- Non-government institutions (the non-governmental mission-driven innovation): the non-government institutions' category in this study groups together the institutions in the

**Table 2. Mission sub-sample to assemble the general mission-driven innovation semantic network.**

| Sector/Continent | # Institutions | % |
|---|---:|---:|
| Government | 24 | 20,0 |
| Asia | 9 | 7,5 |
| Europe | 11 | 9,2 |
| LATAM-CAR | 1 | 0,8 |
| North America | 3 | 2,5 |
| Health | 24 | 20,0 |
| Asia | 5 | 4,2 |
| Europe | 8 | 6,7 |
| North America | 9 | 7,5 |
| Oceania | 2 | 1,7 |
| Higher ed. | 24 | 20,0 |
| Africa | 2 | 1,7 |
| Asia | 11 | 9,2 |
| Europe | 7 | 5,8 |
| North America | 4 | 3,3 |
| Others | 24 | 20,0 |
| Asia | 1 | 0,8 |
| Europe | 5 | 4,2 |
| North America | 18 | 15,0 |
| Private | 24 | 20,0 |
| Asia | 6 | 5,0 |
| Europe | 6 | 5,0 |
| North America | 11 | 9,2 |
| Oceania | 1 | 0,8 |
| Total general | 120 | 100,0 |

Source: the authors' work, based on the organizations' website. Note: LATAM-CAR is Latin America and the Caribbean.

higher education, health, private, and non-profit sectors. In order to explore similarities with the governmental institutions, a sub-sample of the remaining sectors will be assembled. Since both 'Others' and private categories are the least numerous, both will be merged into one category: 'mixed,' composed of 57 institutions. A stratified sample will be implemented to source 57 missions randomly from the higher education and health sectors for a total sub-sample of 171 missions. ($n = 171$—Table 3).

The following permanent link provides access to the missions sourced, thereby facilitating further replications/triangulations [27, 28]:

- https://doi.org/10.34848/ZTL686

## Methods

We will explore the content of the missions via three content analysis techniques: i) sentiment analysis, readability, and lexical diversity; ii) semantic networks; and iii) similarity computation between document corpus.

**Table 3. Mission sub-sample to assemble non-government mission-driven innovation semantic network.**

| Sector/Continent | # Institutions | % |
|---|---|---|
| Health | 57 | 33,3 |
| Asia | 10 | 5,8 |
| Europe | 11 | 6,4 |
| LATAM-CAR | 1 | 0,6 |
| North America | 28 | 16,4 |
| Oceania | 7 | 4,1 |
| Higher ed. | 57 | 33,3 |
| Africa | 1 | 0,6 |
| Asia | 25 | 14,6 |
| Europe | 15 | 8,8 |
| North America | 14 | 8,2 |
| Oceania | 2 | 1,2 |
| Mixed | 57 | 33,3 |
| Asia | 7 | 4,1 |
| Europe | 14 | 8,2 |
| North America | 35 | 20,5 |
| Oceania | 1 | 0,6 |
| Total general | 171 | 100,0 |

Source: the authors' work, based on the organizations' website. Note: LATAM-CAR is Latin America and the Caribbean.

First, while there are limitations in the automated text analysis in terms of identifying whether a mission is expressing its purpose in a *bold* or *ambitious* way and with a *clear* direction, other indices and dictionaries are available that can provide an approximate estimation thereof. A clear and ambitious mission should be expressed in a rich and diverse language but free of unnecessary complexities. Texts with lower readability have been shown to be associated with greater dispersion, lower accuracy, and greater overall uncertainty in analyst earnings forecasts [29, 30]. It has also been found that missions and article abstracts of more reputable journals are articulated with a higher lexical diversity than those of less reputable journals [31, 32].

The equation for the Flesch-Kincaid grade level (FKGL)—one of the most widely used indices to estimate the readability of a text [33]—is:

$$FKGL = 0.39\left(\frac{words}{sentences}\right) + 11.8\left(\frac{syllables}{words}\right) - 15.59$$

Source: Kincaid et al. [34].

The FKGL approximates the equivalent American school grade needed to comprehend any given text at first reading [34]. To analyze the lexical diversity of missions, we used Yule's K [35, 36]. The equation for calculating Yule's K is:

$$K = 10^4 \times \left[-\frac{1}{N} + \sum_{i=1}^{V} f_v(i, N)\left(\frac{i}{N}\right)^2\right]$$

Source: Yule [35].

Where *N* refers to the total number of tokens (i.e., "this or that word on a single line of a single page of a single copy of a book" [37]), *V* to the number of types (i.e., unique tokens),

**Table 4. Scales/Dimensions, number of words, and sample of words by dictionary.**

| Scales/Dimensions | # Words | Sample of words |
|---|---|---|
| Lexicoder Sentiment Dictionary ||| 
| Negative | 2,337 | *termination, discontinued, penalties, misconduct, serious, noncompliance, deterioration, felony* |
| Positive | 353 | *achieve, attain, efficient, improve, profitable* |
| Corporate Social Responsibility content analytic dictionary |||
| Employee | 319 | *adopted child, health benefits, educate, employed, discriminatory* |
| Human Rights | 297 | *aboriginals, fairness, oppressive regime, same-sex, religious diversities* |
| Environment | 451 | *acid rain, conservation, fossils, green engineering, renewable energy* |
| Social and Community | 174 | *transparent, foodbank, indigenous people, social issue* |

Source: the author based on Pencle & Mălăescu; and Young & Soroka [38, 39].

and $f_v(i, N)$ to the number of types occurring *i* times in a sample of length *N*. The lower the results, the fewer repeated words and greater lexical diversity in given text.

We used sentiment analysis to identify related mission sentiment, such as *inspirational*, *bold*, or *ambitious*, by recourse to the Lexicoder Sentiment Dictionary—a lexicon of positive and negative sentiment words [38]. We also used the Corporate Social Responsibility (CSR) content analytic dictionary to identify the reference to topics related to *societal relevance*; the inclusion of *cross-sectional/actors*; and, in broad terms, the involvement of a stakeholder agenda [39]. Both dictionaries aim to capture text content topics related to news, legislative debates, policy documents, and initial public offerings. Experts have validated both dictionaries [38, 39]. Table 4 displays both dictionaries' scales/dimensions, number of words, and examples. Table 5 shows three examples of missions from different sectors with the average word count: ~75, plus respective readability and lexical diversity. It also shows the ratio between positive and negative scales divided by missions' total word count; and the ratio between employee, human rights, environment, and social and community dimensions divided by missions' total word count.

Second, we used semantic networks to study the lexical structure of missions. This enables us to formally assemble the relationship between words and their shared meaning [41]. We conducted the following process to assemble the missions' semantic networks:

- Missions were turned into tokens, and punctuation, stop-words, and non-informative words (e.g., mission(s), aim(s), institution(s), university(ies)) were removed.

- Tokens' co-occurrence was established via a co-occurrence matrix.

- Based on the co-occurrences, a directed-weighted semantic network was assembled.

- We clustered terms using Blondel et al. [42] modularity appraisal.

- We also identified the relevant words via the calculation of centrality, the equation for which is:

$$C_B(p_k) = \sum_{i<j}^{n} \frac{g_{ij}(p_k)}{g_{ij}}; i \neq j \neq k$$

Source: Opsahl et al. [43].

Where $g_{ij}$ is the shorter path that links nodes $p_i$ and $g_{ij}(p_k)$ is the shorter path that links nodes

**Table 5. Example missions and content analysis indicators.**

| Institution | Continent | Sector | Mission | Read. | Div. | Pos. | Neg. | Emp. | Env. | HR | Soc. |
|---|---|---|---|---|---|---|---|---|---|---|---|
| King Mongkut's Institute of Technology Ladkrabang | Asia | Higher ed. | Missions of the Institute's Act consist of 4 categories. 1. Provision of higher education in science and technology of the highest quality toward international standards with good morality. 2. Advancement of knowledge and research in science, engineering, and technology to support the sustainable development of the nation and to ward international excellence 3. Provision of knowledge and innovation for the best academic and Community services. 4. Preservation and promotion of Thai Arts and Culture. | 11,6 | 294,7 | 0 | 12,8 | 9,3 | 10,7 | 5,3 | 9,3 |
| Jiangsu Academy of Agricultural Sciences | Asia | Government | JAAS is one of the largest and earliest comprehensive agricultural/veterinary medicine research institutions in China. It is financed and directly administered by Jiangsu provincial government. Aiming at rural economy and technology development in Jiangsu andChina, the JAAS commitment, including basic and applied research, as well as the extension service, is to advance the production of food and fiber, to protect our natural resources, and to improve the quality of life of all Chinese people. | 16,9 | 251,8 | 0 | 2,2 | 5,3 | 8,0 | 2,7 | 8,0 |
| Santa Fe Institute | North America | Others | Our researchers endeavor is to understand and unify the underlying, shared patterns in complex physical, biological, social, cultural, technological, and even possible astrobiological worlds. Our global research network of scholars spans borders, departments, and disciplines, unifying curious minds steeped in rigorous logical, mathematical, and computational reasoning. As we reveal the unseen mechanisms and processes that shape these evolving worlds, we seek to use this understanding to promote the wellbeing of humankind and of life on earth. | 18,7 | 212,4 | 0 | 0 | 4,0 | 2,7 | 1,3 | 1,3 |

Source: the author based on Pencle & Mălăescu; and Young & Soroka [38, 39] and processed with quanteda [40].

$p_i$ and $p_j p_k$. The higher the value, the higher its betweenness. Betweenness centrality is formally defined as the number of shortest paths that pass through a given node. A node with high betweenness can determine what information gets passed on to another community (i.e., cluster) [44].

- We used a circular layout algorithm [45].

Third, we conducted the following process to compute similarities between mission corpora by sector and continent:

- We prepared the mission texts.

- We then constructed a document-feature matrix. This matrix describes the frequency of terms occurring in each mission text. Terms with ten or fewer appearances were excluded.

- We used the cosine as a similarity distance metric between vectors (i.e., missions converted into term-frequency vectors) [46]. The cosine equation is:

$$\cos \theta = \frac{x \cdot y}{|x||y|}$$

Source: Singhal [46].

Where $x$ and $y$ are two vectors to be compared. The cosine similarity is a number between 0

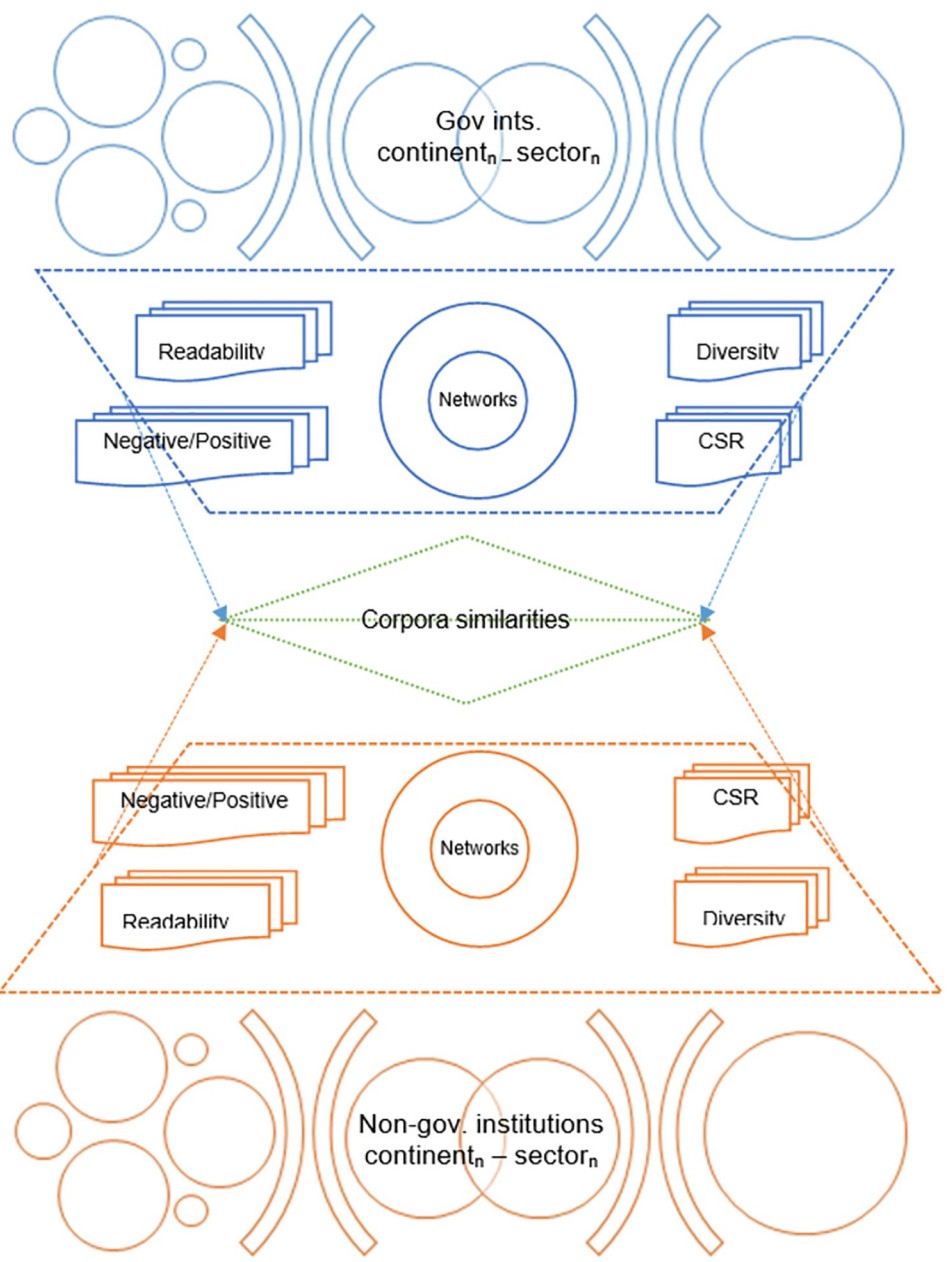

**Fig 1. Graphical summary.** Source: the author.

and 1, with 1 the highest similarity between texts.

- We then plotted a dendrogram based on a hierarchical cluster analysis by sector and continent according to the previous results.

Fig 1 presents a graphical summary of the implemented methodology.

## Software

The software used was quanteda (i.e, content analysis) and igraph (i.e., network analysis) packages for R language, and Gephi (i.e., network analysis and layouts) [40, 45, 47, 48].

## Results

Table 6 presents the average of the analyzed variables: readability, diversity, and dictionary term ratios by sector and continent. A frequently asked question by practitioners is: How long should a mission be? The average word count is ~75 words. There are missions as long as 1160 + words; others are as concrete as *'To Serve Patients'* (Amgen) or *'To promote health'* (Kuopio University Hospital). Between sectors, the private was the sector with the lowest word average:

**Table 6. Averages of readability, diversity, and dictionary term ratios by sector and continent.**

| Sector/Continent | Word count Av. | Av. Readability | Av. Diversity | Av. Employee ratio | Av. Environment ratio | Av. Human rights ratio | Av. Social and community ratio | Av. Negative ratio | Av. Positive ratio |
|---|---|---|---|---|---|---|---|---|---|
| Government | 80,8 | 19,9 | 523,4 | 4,8 | 7,8 | 2,4 | 5,5 | 1,2 | 4,8 |
| Africa | 106,7 | 20,9 | 435,7 | 4,2 | 11,4 | 2,0 | 8,1 | 1,0 | 6,8 |
| Asia | 82,5 | 20,9 | 554,0 | 5,1 | 8,4 | 2,9 | 5,6 | 1,1 | 4,5 |
| Europe | 89,8 | 19,8 | 442,0 | 4,3 | 7,0 | 2,1 | 4,9 | 1,0 | 4,5 |
| LATAM-CAR | 45,0 | 26,5 | 533,2 | 8,6 | 7,7 | 6,8 | 7,7 | 1,5 | 6,0 |
| North America | 55,3 | 18,1 | 675,7 | 4,9 | 7,6 | 2,2 | 6,0 | 2,1 | 5,6 |
| Oceania | 38,7 | 17,1 | 696,9 | 7,0 | 10,7 | 2,7 | 8,3 | 0,5 | 7,7 |
| Health | 51,5 | 15,8 | 780,0 | 11,8 | 6,7 | 6,5 | 8,1 | 1,1 | 8,5 |
| Africa | 25,4 | 15,3 | 771,5 | 15,4 | 14,5 | 5,1 | 10,1 | 0,0 | 10,2 |
| Asia | 66,2 | 16,9 | 590,8 | 11,6 | 7,1 | 6,6 | 8,2 | 1,3 | 7,6 |
| Europe | 52,1 | 14,8 | 813,7 | 9,4 | 5,3 | 5,4 | 6,6 | 1,0 | 7,7 |
| LATAM-CAR | 65,0 | 13,9 | 324,1 | 12,3 | 6,2 | 3,1 | 3,1 | 0,0 | 5,6 |
| North America | 49,7 | 16,3 | 787,6 | 12,8 | 6,8 | 7,1 | 8,2 | 1,0 | 9,2 |
| Oceania | 31,5 | 14,2 | 1045,9 | 13,5 | 8,7 | 7,2 | 11,7 | 1,8 | 8,7 |
| Higher ed. | 81,1 | 21,0 | 451,5 | 8,2 | 6,7 | 4,7 | 8,1 | 0,8 | 6,6 |
| Africa | 70,1 | 22,9 | 420,3 | 9,7 | 7,3 | 5,9 | 9,7 | 1,0 | 7,4 |
| Asia | 77,0 | 20,9 | 447,7 | 8,1 | 6,6 | 4,6 | 7,7 | 0,7 | 6,1 |
| Europe | 90,6 | 21,6 | 465,2 | 6,8 | 6,6 | 4,0 | 6,9 | 0,8 | 6,0 |
| LATAM-CAR | 61,2 | 23,6 | 511,7 | 9,0 | 6,6 | 4,8 | 6,6 | 1,4 | 3,4 |
| North America | 80,0 | 20,4 | 424,5 | 9,4 | 6,7 | 5,4 | 9,4 | 1,0 | 7,6 |
| Oceania | 69,4 | 19,1 | 677,2 | 10,2 | 8,4 | 6,2 | 11,9 | 0,3 | 6,7 |
| Others | 64,3 | 17,9 | 498,1 | 5,5 | 7,7 | 3,5 | 6,5 | 1,4 | 6,3 |
| Asia | 16,0 | 11,3 | 1111,1 | 6,3 | 12,5 | 0,0 | 0,0 | 0,0 | 0,0 |
| Europe | 114,2 | 16,5 | 472,6 | 7,8 | 6,3 | 5,9 | 7,2 | 0,5 | 6,9 |
| North America | 53,1 | 18,6 | 471,1 | 4,9 | 7,8 | 3,0 | 6,6 | 1,7 | 6,5 |
| Private | 47,4 | 13,4 | 890,7 | 5,9 | 4,1 | 4,1 | 8,4 | 1,8 | 7,9 |
| Asia | 67,3 | 16,1 | 563,5 | 5,6 | 2,8 | 3,8 | 7,5 | 0,0 | 4,2 |
| Europe | 39,7 | 10,7 | 722,8 | 4,5 | 4,0 | 3,0 | 8,6 | 0,0 | 10,2 |
| North America | 45,4 | 13,8 | 1110,7 | 6,4 | 3,8 | 4,8 | 8,8 | 3,2 | 8,2 |
| Oceania | 31,0 | 15,9 | 625,0 | 12,9 | 16,1 | 3,2 | 3,2 | 6,2 | 6,2 |
| Total general | 75,0 | 19,8 | 528,3 | 8,2 | 6,8 | 4,7 | 7,7 | 1,0 | 6,6 |

Source: the author based on Pencle & Mălăescu; and Young & Soroka [38, 39] and processed with quanteda [40].

~47; higher education institutions, on the other hand, showed the highest average: ~81. Regarding regions, Europe-Other had the highest word average: ~114; while Oceania-private had the lowest: ~31. The overall average of the FKGL was ~20; the sector and continent missions with the highest readability were private with ~13 and Europe with ~10, respectively. The overall average of Yule's K was ~528; the sector and continent missions with the highest diversity were higher education with ~451 and Latin America and the Caribbean (LATAM-CAR) with ~324, respectively.

The overall average of the negative words and mission word count ratio was ~1 (100 as the highest ratio); the sector and continent missions with the highest ratio were private with ~1 and Oceania with ~6, respectively. The overall average of the positive words and mission word count ratio was ~6; the sector and continent missions with the highest ratio were health with ~8 and Africa and Europe with ~10, respectively. The overall average of the employee words dimension and mission word count ratio was ~8; the sector and continent missions with the highest ratio were health with ~11 and Oceania with ~12, respectively. The overall average of the environment words dimension and mission word count ratio was ~6; the sector and continent missions with the highest ratio were government with ~7 and Oceania with ~16, respectively. The overall average of the human rights words dimension and mission word count ratio was ~4; the sector and continent missions with the highest ratio were health with ~6 and North America with ~7, respectively. The overall average of the social and community words dimension and mission word count ratio was ~8; the sector and continent missions with the highest ratio were health and higher education with ~8 and Oceania with ~11, respectively.

Fig 2 presents correlations and distribution plots for the variables examined for the complete sample. Since most of the correlations were significant $p \leq .05$ (*), we only discuss those with an $r \geq .7$ and $p \leq .001$ (***). There is a coherent higher and significant correlation between word count and diversity, particularly in the 'Others' sector. Shorter missions (fewer

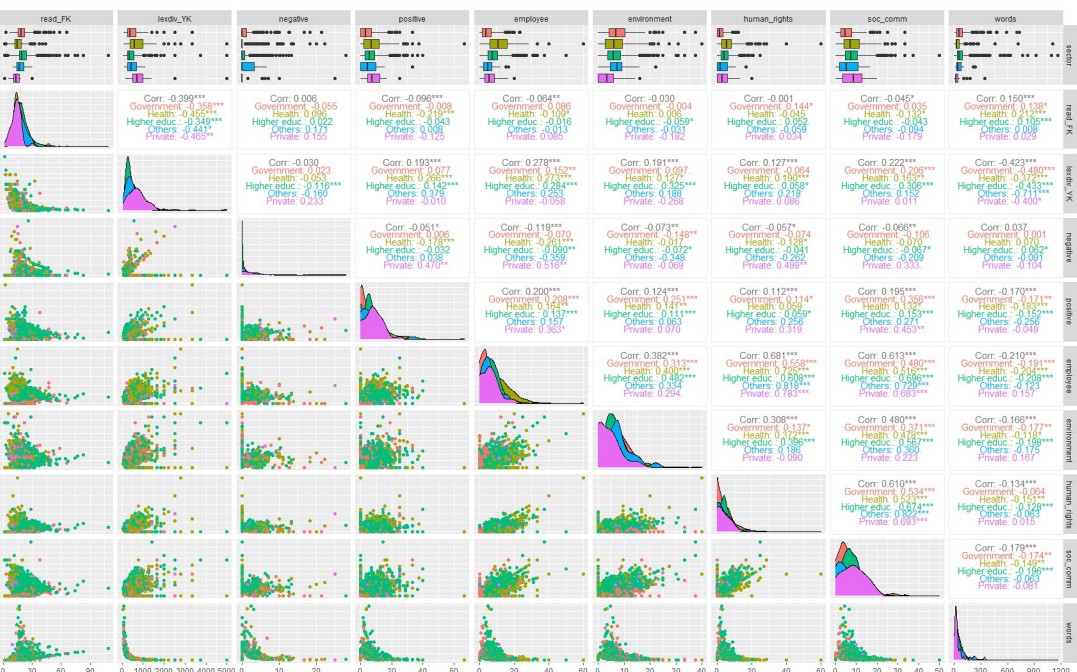

**Fig 2. Correlation and distribution plots for readability, lexical diversity indices, and dictionary terms and total word ratio by sector.** Source: the author based on Pencle & Mălăescu; and Young & Soroka [38, 39] and processed with quanteda [40].

words), have the lowest diversity (higher Yule's K). There are higher and significant correlations between words from the Corporate Social Responsibility content analytic dictionary. Consequently:

- Dimensions concerned with employees and human rights are mutually present in the missions of 'Others' and private sectors.

- Dimensions concerned with employees and social-community issues are mutually present in the 'Others' sector missions.

- Dimensions concerned with human rights and social-community issues are also mutually present in the 'Others' sector missions.

Figs 3–5 display the semantic networks. The top-four clusters were colored in yellow (principal component), aqua, green, and fuchsia. Clusters were labeled manually based on nodes' interrelatedness. Only the top-five nodes (i.e., words) with the highest betweenness of each

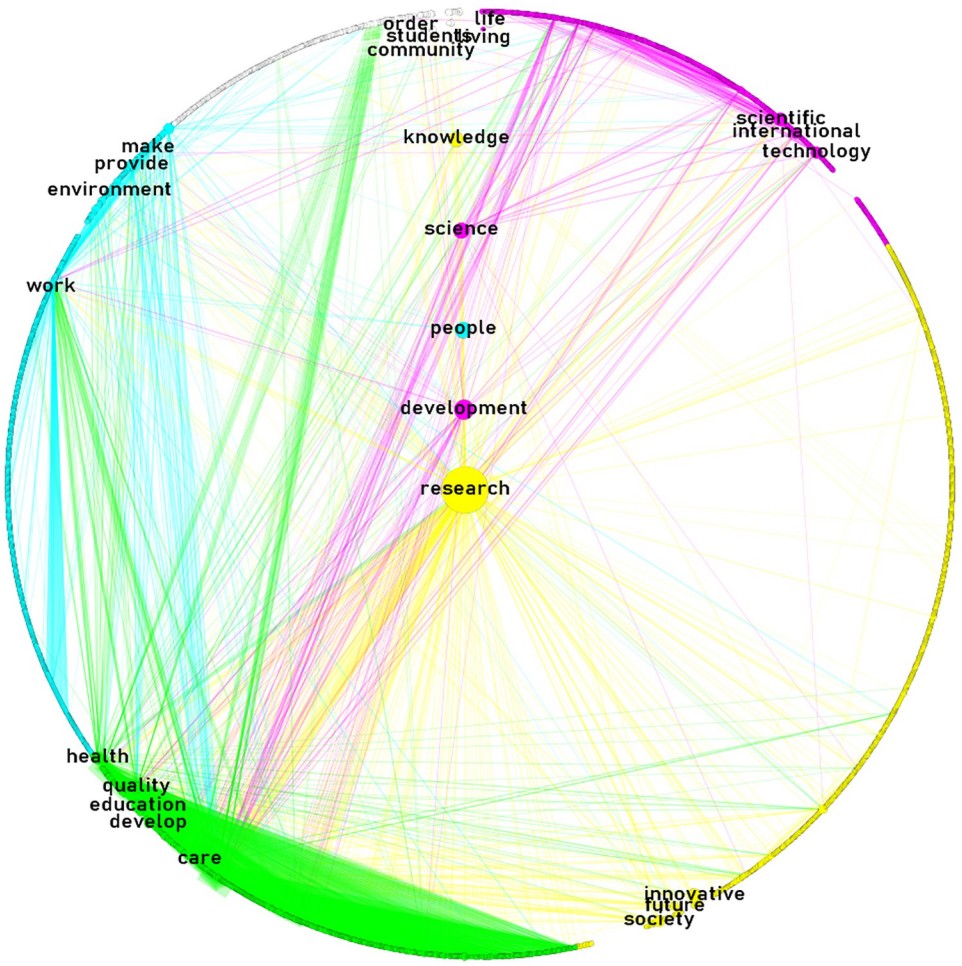

**Fig 3. The general mission-driven innovation semantic network.** Source: the author's processed with Gephi, quanteda, and igraph [40, 45, 47, 48]. Notes: network layout: circular; links visible: 2.46% (link weight range 10–576); link color: node of origin; node size proportional to betweenness centrality score; nodes labeled: top-five of highest betweenness score of each cluster.

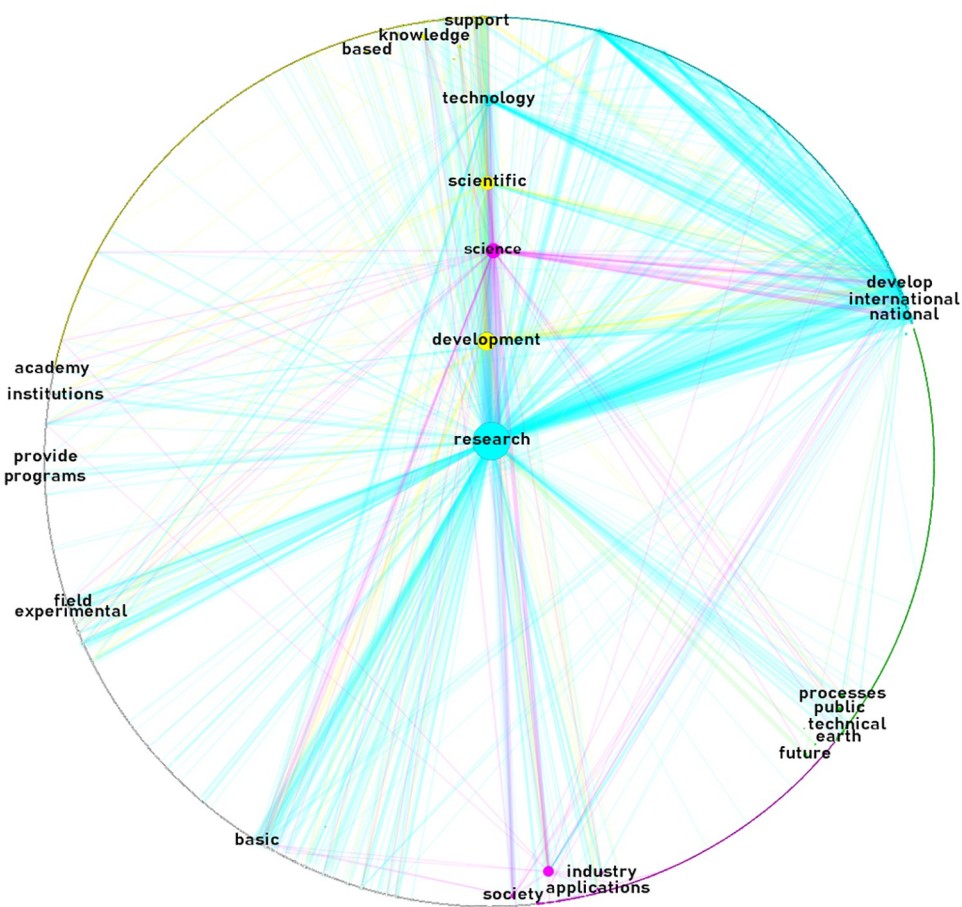

**Fig 4. The governmental mission-driven innovation semantic network.** Source: the author's processed with Gephi, quanteda, and igraph [40, 45, 47, 48]. Notes: network layout: circular; links visible: 0.31% (link weight range 25–807); link color: node of origin; node size proportional to betweenness centrality score; nodes labeled: top-five of highest betweenness score of each cluster.

cluster were labeled. Nodes at the center are the top-five betweenness nodes in the overall network.

Fig 3 displays the general mission-driven innovation semantic network. It is composed of 2,130 nodes connected by 189,000+ links. The main four clusters grouped 88% of the nodes; 31% (research for society's future), 24% (human capital), 17% (health education/learning), and 16% (international science, technology and innovation), respectively. The role of research activities and their results for society's future is noteworthy. Most of the words with higher betweenness belong to activities (e.g., R&D) and the development of science and knowledge, but also to the role of people involved in the process.

Fig 4 displays the governmental mission-driven innovation semantic network. It is composed of 3,999 nodes connected by 423,000+ links. The main four clusters grouped 71% of the nodes; 22% (academic R&D), 20% (glocal—local and global—R&D), 16% (public issues), and 13% (applied R&D), respectively. In contrast with the general mission-driven semantic network, where there is an explicit mention of people's role (Fig 3), the governmental mission-driven semantic network shows a higher betweenness for activities related to R&D and science. The network also shows the relevance not only of the internationalization of STI but also its

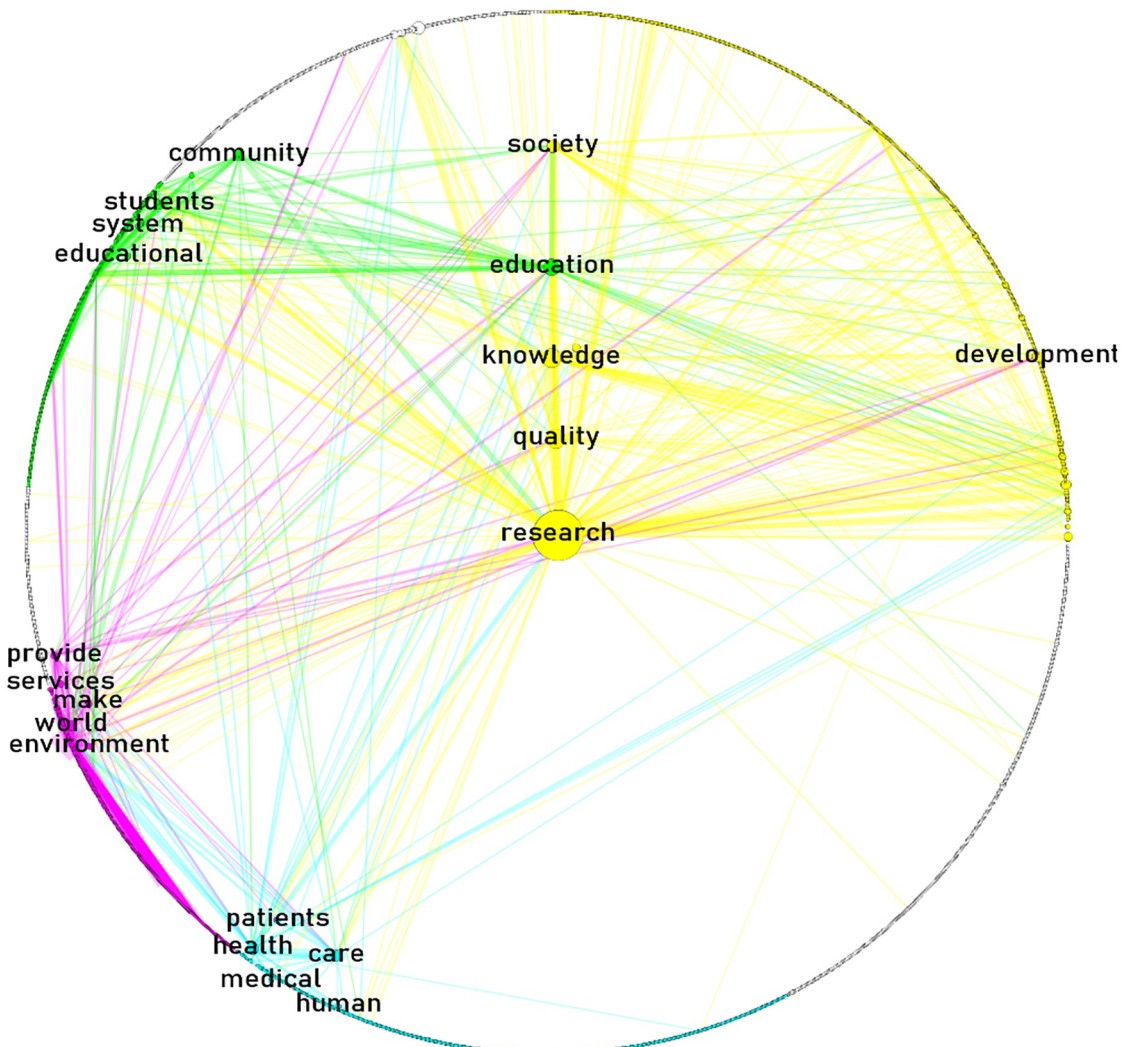

**Fig 5. Non-government mission-driven innovation semantic network: Higher education, health, private and other institutions.**
Source: the author's processed with Gephi, quanteda, and igraph [40, 45, 47, 48]. Notes: network layout: circular; links visible: 1.09% (link weight range 10–231); link color: node of origin; node size proportional to betweenness centrality score; nodes labeled: top-five of highest betweenness score of each cluster.

relevance in the local context; a concern with public issues; and a tacit division between academic and applied R&D clusters.

Fig 5 displays the non-government mission-driven innovation semantic network. It is composed of 2,089 nodes connected by 112,209+ links. The main four clusters grouped 63% of the nodes; 25% (basic/applied R&D), 18% (health services/stakeholders), 10% (higher ed. stakeholders/outreach), and 10% (environmental products/services), respectively. Compared to the previous two networks, the non-government mission-driven innovation semantic network shows a higher betweenness for terms related to quality, society, and education; basic and applied clusters merge into one; and there is a central role of health and environmental products/services, and outreach and stakeholder in higher education. The common word with the highest betweenness in all the networks was: research. This word, however, belongs to different clusters in each network: research for society's future; research in a glocal contexts; and research in both basic/applied contexts.

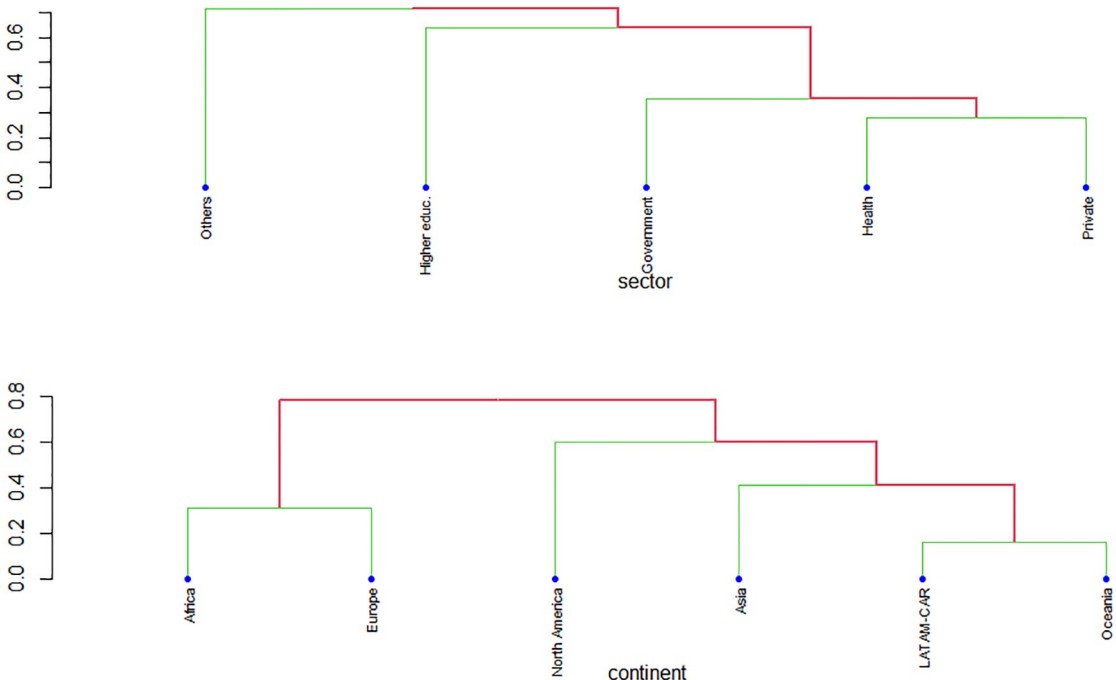

**Fig 6. Cosine similarity dendrograms by sector (top) and continent (bottom).** Based on the all sector stratified sample n = 120. Source: the author's and processed with quanteda [40, 47].

Fig 6 displays the similarity dendrograms by sector (top) and continent (bottom) applied to the general mission-driven innovation sample. The sector dendrogram shows that the 'Others' sector missions are a *simplicifolious* clade (i.e., separate/different from other clades). The other chunks are clustered into groups: there is a greater similarity between health and private sector missions than between higher education and government missions; and a greater similarity between government missions and the missions of health and private than higher education missions. On the other hand, the continent dendrogram displays similarities between the missions of LATAM-CAR and Oceania, and, in a different clade, between Africa and Europe. Asia's missions are similar to those of LATAM-Oceania; and there is a greater similarity between North America's missions and those of Asia than between North America's missions and those of Africa-Europe.

## Discussion and concluding remarks

Regarding our RQ1: *What are the content and its structure of the missions of knowledge-intensive institutions*? Missions are challenging to read texts with lower lexical diversity, reaching the readability level of academic papers or specialized business columns, but of inferior lexical diversity compared to that of a Harry Potter novel [33, 49, 50].

Despite the decreasing readability in the academic literature—recalling the dominant presence of higher education institutions in the overall sample—the average FKGL computed here is similar to that of missions from Latin American companies and management journals [9, 31, 51]. Lexical diversity is vastly inferior to that of other related strategic texts, such as management journal missions and abstracts [31, 32]. Accordingly, practitioners should focus on increasing the readability and lexical diversity of organization missions. Ongoing research has shown that missions with positive content, i.e., a higher ratio of positive words to negative

words, belong to higher ranking institutions [52]. Similarly, controversial research in social media has shown evidence of emotional contagion when users are exposed to more or less positive content [53]. Our results, however, do not provide conclusive evidence of a link between mission positive content and emotional contagion within institutions; further research is needed to justify such an assertion. The case of Oceania deserves a closer look since it surpassed the average ratio in the employee, environment, and social-community dimensions. Within the parameters of our study, this region comprised two countries, Australia and New Zealand, in respect of which most of the missions belonged to higher education and health institutions. A study on leading companies in Australia and other countries—all members of the Global Compact—found an effect on environmental and workers factors in their CSR reporting [54].

A predictable outcome was the higher and significant correlation between employee and human rights/social community; and human rights and social-community dimensions in the 'Others' (i.e., non-profit) sector. To the best of our knowledge, we are the first to confirm this higher and significant correlation in non-profit institutions since most of the studies that use the Corporate Social Responsibility content analytic dictionary focus on the private sector [39, 55, 56]. A study conducted in microfinance organizations showed a strong correlation between missions that highlight actions tackling poverty, women's empowerment, and financial inclusion and actual practice [11]. Consequently, our study shows that non-profit sector missions integrate a higher ratio of several correlated Corporate Social Responsibility words, particularly those from the employee, human rights, and social community dimensions, which relate to the *societal relevance* and to the inclusion of *cross-sectional/actors* mission components. As noted above, there are multiple significant correlations between other content variables that remain undiscussed in the literature and could be the focus of future studies.

Regarding our RQ$_2$. *Are there similarities between the content and its structure of government and non-government missions*? 'Research' was a common word co-occurring with the highest betweenness in all networks. It belonged, however, to different clusters in each network: research for society's future, in the general network; research in a glocal context, in the government network; and research in both basic/applied contexts, in the non-governmental network. Although the higher betweenness and frequency of 'research' in the missions of research-knowledge intensive institutions is nothing new [6, 52], we show the changing context of the same word in different sectors. Previous research has pointed out that the mission clusters of research-knowledge intensive institutions are mainly focused on identifying the principal products and services, and the specification of key elements in the institution's philosophy [52, 57], which is also consistent with our findings. For instance, we found that the principal components of the mission clusters of research-knowledge intensive institutions were related to research for society's future (institution's philosophy); and academic/basic-applied R&D (principal services).

On the other hand, clusters observed in our study that have not been detected in previous exercises focused on research-knowledge intensive institutions [52], are those related to geographic domain or influence, as reflected in the international science, technology and innovation; and glocal R&D clusters. The three networks here examined shared the common thread of 'research,' although it changed in according to each cluster's composition. There are also other sector-specific emphases, such as the glocal R&D or public issues clusters in the government missions, compared to those of higher education stakeholders/outreach or the health services/stakeholders of the non-government missions. Viewed from that perspective, network structures among government and non-government missions are different in terms of clusters. Further research could improve and refine the method used for labeling clusters.

Lastly, regarding our RQ$_3$: *Are mission content similarities identifiable across sectors and continents*? The similarities between mission corpus across sectors revealed the independence

of 'Others'—previously discussed—and clustering between health and private sectors. This could be contrasted with the use of particular managerial mission terms among hospitals and health care institutions (e.g., distinctive competence/strengths or concern for satisfying patients/customers/employees) and their relationship to higher performance [58–60]. Regarding continent similarities, there was a clustering between LATAM-CAR and Oceania, and between Africa and Europe. A feature to highlight is the clustering between higher and middle-income regions. This could be supporting evidence for the *isomorphism* observed in the mission statement of universities [6]; and for the influence of the mission statement as a conceptual framework for strategic planning, such as that utilized by the USA towards Europe, and then to other regions [3, 61, 62]. In other words, particular corpora similarities between sectors and continents might be explained by institutions in higher-income countries that use the mission statement as part of their strategic planning progressively influencing other regions. Further research could use other distance measurements.

We outline the implications of our results in two lines regarding research and practice. First, our open-access dataset on missions worldwide can be used as a source for further replication, triangulation, or crowdsourcing-data studies in which the strategic planning tools are being assessed or reused for further purposes (e.g., strategic planning and performance related-research), therefore contributing to efforts in conducting a more open, cumulative, and edifying research [27, 28, 63]. Second, at the practice level, strategic planning activities usually begin with developing a benchmarking reporting, looking for a comparison, similarities, and differentiation with other organizations of a similar sector. Since strategic planning activities could demand between 5–6 months [64], our open-access dataset and insights here presented and discussed could substantially facilitate practitioners' work.

In conclusion, this study presented the similarities and differences in the mission content of a worldwide sample of cross-sectorial research-knowledge intensive institutions through the deployment of sentiment analysis, readability, and lexical diversity; semantic networks; and a similarity computation between document corpus. We argued—and in most parts, confirmed —that: (i) missions are challenging to read texts with lower lexical diversity that avoid the use of negative words and favor the use of positive ones; (ii) that the non-profit sector missions share multiple dimensions in the use of CSR jargon; (iii) that 'research' changed according to mission sectorial context and that each sector has a cluster-specific focus; and (iv) that sectors and continents both share mission corpora similarities that might be explained by the use of the mission as a strategic planning tool.

Further studies could design automated methodologies for optimizing the readability, diversity, and sentimental content of missions as a real-time tool used during strategic planning exercises; could explore the significant correlations between remaining content variables; and improve the labeling of clusters. Also, further research efforts on the creation, validation, and inclusion of non-English languages and dictionaries could incorporate missions from other regions, thereby increasing the research scope. Finally, but most importantly, future studies could establish the correlation between mission content and how institutions execute their functions (e.g., Has a higher ratio of positive words in an institution's mission have any bearing on it being a *Great Place to Work*? Or: Does a higher human rights dimension ratio translate into a successful human rights profile as assessed by communities and local and national governments?).

## Acknowledgments

The authors thank Dr. Francesca Cauchi for editing an early draft of this manuscript.

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
