## [Decision Letter · Decision Letter 0]

26 Mar 2022

PONE-D-21-37282What is the mission of innovation?PLOS ONE

Dear Dr. Cortes,

Thank you for submitting your manuscript to PLOS ONE. After careful consideration, we feel that it has merit but does not fully meet PLOS ONE’s publication criteria as it currently stands. Therefore, we invite you to submit a revised version of the manuscript that addresses the points raised during the review process.In your revision you will be extremely careful in addressing all the point, especially the one below:The article aims to analyse missions but there is a fundamental error: the article confuses mission-oriented polices and organisational missions. The analysis is actually based on the latter but the relevance comes from the former. There is some minimal overlap, but in theory and practice these are two very different phenomena that the author fails to understand.

We look forward to receiving your revised manuscript.

Kind regards,

Rosella Levaggi

Academic Editor

PLOS ONE

Journal Requirements:

Dear Authors

The reviewers have found mertis and flaws in your article.

I would like to give a chiance to revise your paper, provided that you can change your contribution so to take into account this problem:

"The article aims to analyse missions but there is a fundamental error: the article confuses mission-oriented polices and organisational missions. The analysis is actually based on the latter but the relevance comes from the former. There is some minimal overlap, but in theory and practice these are two very different phenomena that the author fails to understand."

Reviewers' comments:

Reviewer's Responses to Questions

**Comments to the Author**

1. Is the manuscript technically sound, and do the data support the conclusions?

Reviewer #1: Partly

Reviewer #2: Yes

2. Has the statistical analysis been performed appropriately and rigorously? 

Reviewer #1: Yes

Reviewer #2: Yes

3. Have the authors made all data underlying the findings in their manuscript fully available?

Reviewer #1: Yes

Reviewer #2: No

4. Is the manuscript presented in an intelligible fashion and written in standard English?

Reviewer #1: Yes

Reviewer #2: Yes

5. Review Comments to the Author

Reviewer #1: The article aims to analyse missions but there is a fundamental error: the article confuses mission-oriented polices and organisational missions. The analysis is actually based on the latter but the relevance comes from the former. There is some minimal overlap, but in theory and practice these are two very different phenomena that the author fails to understand.

Reviewer #2: Dear Author,

I recommend to review the title of the research, according to its content. The question in the title is too general, without an obvious connection with the analysis, and the reader would expect some answers that are not given in the study. The title asks “What is the mission of innovation?” and the answer is “In conclusion, this study presented the similarities and differences in the mission content of a worldwide sample of cross-sectorial research-knowledge intensive institutions through the deployment of sentiment analysis, readability, and lexical diversity.”

At the same time, the abstract should be better systematized, pointing out more clearly what is the research aims and what results it has reached. Also, discussing the implications of these results would bring greater value to the paper.

6. PLOS authors have the option to publish the peer review history of their article (what does this mean?). If published, this will include your full peer review and any attached files.

Reviewer #1: No

Reviewer #2: No

---

## [Author Response · Author response to Decision Letter 0]

28 Mar 2022

Dear editor, reviewers—

The authors appreciate immensely the comments and suggestions for improving the article. Please, find attached the response to each comment according to each reviewer. Also, the manuscript with track changes and clean. 

Sincerely, 

The authors.

---

## [Decision Letter · Decision Letter 1]

11 Apr 2022

What is the mission of innovation? — Lexical structure, sentiment analysis, and cosine similarity of mission statements of research-knowledge intensive institutions

PONE-D-21-37282R1

Dear Dr. Cortes,

We’re pleased to inform you that your manuscript has been judged scientifically suitable for publication and will be formally accepted for publication once it meets all outstanding technical requirements.

Kind regards,

Rosella Levaggi

Academic Editor

PLOS ONE

Additional Editor Comments (optional):

The reviewers are satisfied with your changes and I think that the paper can be published

Reviewers' comments:

Reviewer's Responses to Questions

**Comments to the Author**

1. If the authors have adequately addressed your comments raised in a previous round of review and you feel that this manuscript is now acceptable for publication, you may indicate that here to bypass the “Comments to the Author” section, enter your conflict of interest statement in the “Confidential to Editor” section, and submit your "Accept" recommendation.

Reviewer #2: All comments have been addressed

2. Is the manuscript technically sound, and do the data support the conclusions?

Reviewer #2: Yes

3. Has the statistical analysis been performed appropriately and rigorously? 

Reviewer #2: Yes

4. Have the authors made all data underlying the findings in their manuscript fully available?

Reviewer #2: Yes

5. Is the manuscript presented in an intelligible fashion and written in standard English?

Reviewer #2: Yes

6. Review Comments to the Author

Reviewer #2: (No Response)

7. PLOS authors have the option to publish the peer review history of their article (what does this mean?). If published, this will include your full peer review and any attached files.

Reviewer #2: No